# Modification of Hydroxypropyltrimethyl Ammonium Chitosan with Organic Acid: Synthesis, Characterization, and Antioxidant Activity

**DOI:** 10.3390/polym12112460

**Published:** 2020-10-23

**Authors:** Yingqi Mi, Wenqiang Tan, Jingjing Zhang, Zhanyong Guo

**Affiliations:** 1Key Laboratory of Coastal Biology and Bioresource Utilization, Yantai Institute of Coastal Zone Research, Chinese Academy of Sciences, Yantai 264003, China; yqmi@yic.ac.cn (Y.M.); wqtan@yic.ac.cn (W.T.); jingjingzhang@yic.ac.cn (J.Z.); 2Center for Ocean Mega-Science, Chinese Academy of Sciences, 7 Nanhai Road, Qingdao 266071, China; 3University of Chinese Academy of Sciences, Beijing 100049, China

**Keywords:** antioxidant activity, hydroxypropyltrimethyl ammonium chloride chitosan, organic acid salt, ion exchange

## Abstract

A novel and green method for the preparation of chitosan derivatives bearing organic acids was reported in this paper. In order to improve the antioxidant activity of chitosan, eight different hydroxypropyltrimethyl ammonium chitosan derivatives were successfully designed and synthesized via introducing of organic acids onto chitosan by mild and non-toxic ion exchange. The data of Fourier Transform Infrared (FTIR), ^13^C Nuclear Magnetic Resonance (NMR), ^1^H NMR, and elemental analysis for chitosan derivatives indicated the successful conjugation of organic acid salt with hydroxypropyltrimethyl ammonium chloride chitosan (HACC). Meanwhile, the antioxidant activity of the chitosan derivatives was evaluated in vitro. The results indicated that the chitosan derivatives possessed dramatic enhancements in DPPH-radical scavenging activity, superoxide-radical scavenging activity, hydroxyl radical scavenging ability, and reducing power. Furthermore, the cytotoxicity of the synthesized compounds was investigated in vitro on L929 cells and showed low cytotoxicity. Thus, the enhanced antioxidant property of all novel chitosan products might be a great advantage, while applied in a wide range of applications in the form of antioxidant in biomedical, food, and cosmetic industry.

## 1. Introduction

Reactive oxygen species (ROS), which contains superoxide anion (·O_2_^−^), hydrogen peroxide (H_2_O_2_), hydroxyl radical (·OH), and singlet oxygen (^1^O_2_), are produced by normal metabolism in vivo, radiation, and chemical factors [1,2,3]. The constant quantity of ROS is of vital importance to maintain the stability of the intracellular environment. Organisms maintain the stability of ROS quantity through the antioxidant defense mechanism in order to achieve the dynamic balance between the challenge of oxidation and protection [4,5]. Under the circumstances of oxidative stress, excessive production of ROS limits the antioxidant defense mechanism, which can lead to aging or disease in the body [6]. Generally speaking, free radicals, one of the most potentially lethal type of reactive oxygen species, may cause a range of injuries in tissues and cells [6,7]. Fortunately, antioxidants can promote health and prevent disease by scavenging free radicals in living organisms. Moreover, antioxidants, as a kind of food preservative, have also obtained more attention because they can prevent foods from decay through oxidation [5]. Nevertheless, the use of synthetic small-molecule antioxidants has been related to health risks, resulting in the rigorous regulation concerning their use in food and pharmaceutical industries [5,8]. In recent years, with the significant improvement consumer consciousness of food safety, seeking natural polysaccharides and their derivatives with antioxidant activity in order to use in the pharmaceutical and food industry has been drawing more attention.

There are abundant marine resources which can be developed into antioxidant alternatives. Chitosan, (1-4)-2-amino-2-deoxy-b-*d*-glucan, is one of the marine bioactive compounds, which can be readily prepared by the *N*-deacetylation from chitin [2,9,10]. In the last few years, chitosan has attracted significant attention in chemical industry, maquillage, papermaking, and food science because of its excellent physical and chemical properties such as biodegradability, non-toxicity, biocompatibility, bioactivity and other properties [11,12]. However, the weak bioactivity and poor water solubility could greatly limit the further industrial application of chitosan. Therefore, the structural and functional modification of chitosan so as to obtain different chitosan derivatives with well water solubility and bioactivity is the research emphasis [13,14]. As a kind of polycationic compound, hydroxypropyltrimethyl ammonium chloride chitosan (HACC) possessed good water solubility, moisture absorption, hypotoxicity, and biocompatibility. Nevertheless, the antioxidant activity of HACC was rarely reported.

In addition, polyphenols possessed multiple bioactivities such as antioxidant, antifungal activity, and antitumor activity, which are abundant secondary metabolites of plants in nature [15]. In particular, some hydroxyl acids, including ascorbic acid, gallic acid, caffeic acid, ferulic acid and *p*-coumaric acid, have been reported to possess excellent antioxidant activity. It is worth noticing that most hydroxyl acids have both carboxyl (–COOH) and hydroxyl (–OH) groups, which can easily react with chitosan. In previous studies, we used halogen compounds to modify chitosan to obtain a series of chitosan derivatives with antifungal activity [9]. So, we consider modifying chitosan with hydroxy acids. When these hydroxyl acids are grafted to chitosan backbone chain, they should be released slowly and increased the antioxidant activity of chitosan.

Based on the above, we hope to obtain a kind of novel chitosan derivatives with good bioactivity, biocompatibility, and degradability, which will be widely used as an antifungal agent or antioxidant in food, medicine, cosmetics, and other fields. Therefore, we modified hydroxypropyltrimethyl ammonium chloride chitosan with eight different organic acid salts via a mild and non-toxic route (ion exchange) to acquire water-soluble and strong antioxidant activity chitosan derivatives, which are coumarate-hydroxypropyltrimethyl ammonium chitosan conjugates (HACCu), ascorbate-hydroxypropyltrimethyl ammonium chitosan conjugates (HACAs), ferulate-hydroxypropyltrimethyl ammonium chitosan conjugates (HACFe), caffeate-hydroxypropyltrimethyl ammonium chitosan conjugates (HACCa), *p*-coumarate-hydroxypropyltrimethyl ammonium chitosan conjugates (HACPc), salicylate-hydroxypropyltrimethyl ammonium chitosan conjugates (HACSa), hydroxybenzoate-hydroxypropyltrimethyl ammonium chitosan conjugates (HACHy), and gallate-hydroxypropyltrimethyl ammonium chitosan conjugates (HACGa), respectively. Therefore, we synthesized hydroxypropyltrimethyl ammonium chloride chitosan firstly. Then, organic acid salt-hydroxypropyltrimethyl ammonium chitosan conjugates were synthesized by ion exchange. The structures of chitosan derivatives were characterized by FTIR, ^1^H Nuclear Magnetic Resonance (NMR), ^13^C NMR, and elemental analyses. Meanwhile, DPPH-radical scavenging activity, superoxide-radical scavenging activity, hydroxyl radical scavenging ability, and reducing power of chitosan derivatives were evaluated in vitro. Besides, the cytotoxicity of the synthesized compounds was investigated in vitro on L929 cells and all the tested compounds showed low cytotoxicity.

## 2. Results and Discussion

### 2.1. Chemical Synthesis and Characterization

The organic acid salt-hydroxypropyltrimethyl ammonium chitosan conjugates were synthesized by multistep reactions (Scheme 1). The structures of chitosan derivatives were characterized by FTIR, ^1^H NMR, ^13^C NMR and elemental analysis.

#### 2.1.1. FTIR Spectra

Figure 1 shows the spectra of chitosan, hydroxypropyltrimethyl ammonium chloride chitosan, and eight different organic acid salt-hydroxypropyltrimethyl ammonium chitosan conjugates, respectively. For chitosan, the peaks appear at 3405 cm^−1^ (angular deformation of O–H and N–H), 2919 cm^−1^ (–CH stretching), and 1628 cm^−1^ (vibration modes of amino group) [16]. As for HACC, the spectrum shows the introduction of the quaternary ammonium salt group. Specifically, the peak of 1478 cm^−1^ shows the stretching vibration of C–H from the trimethylammonium group [17]. Meanwhile, the peak at 1628 cm^−1^ weakens greatly for the partial change of the primary amine of chitosan to the secondary amine. After the introduction of coumarate, the spectrum of HACCu appears with peaks at 1739 cm^−1^, which can be attributed to the C=O stretching vibration of coumarate. After the introduction of ascorbate, the spectrum of HACAs shows the new peaks at 1715, 756 cm^−1^ for the absorption of ascorbate anions. Compared to HACC, new peaks of the spectra of HACFe appear at 1585, 1546 cm^−1^, which were assigned to the characteristic peaks of the benzene ring. As for HACHy, HACSa, HACPc, HACCa, and HACGa, new peaks appear at 1587, 1596, 1596, 1545, and 1552 cm^−1^, respectively, can be attributed to the C–C stretching vibration of benzene ring [3,18]. Furthermore, for the spectrum of HACSa, HACCa, HACPc, and HACGa, absorption peaks at 781, 767, 772, 735, 739 cm^−1^ corresponded to the C–C stretching vibration of the benzene ring [19]. Meanwhile, the peak at 1478cm^−1^ assigned to the C–H bending the trimethylammonium group still exists in the molecules of HACCu, HACAs, HACFe, HACHy, HACSa, HACPc, HACCa, and HACGa. Hence, the data preliminarily confirmed the successful conjugations of organic acid salt and hydroxypropyltrimethyl ammonium chitosan.

#### 2.1.2. NMR Spectra

The ^1^H NMR spectra of chitosan and chitosan derivatives are shown in Figure 2. Chemical shifts at δ 4.48 ppm, δ 3.21–4.08 ppm, and δ 2.69 ppm are attributed to [H1], [H3]–[H6], and [H2] of chitosan. For the spectrum of HACC, a new chemical shift (δ 3.18 ppm) appeared, which can be assigned to hydrogen of trimethyl ammonium groups. Meanwhile, the new characteristic peaks are observed easily from the ^1^H NMR spectra of HACC: δ 3.34 ppm attributed to –CH_2_, δ 2.71 ppm (a), δ 4.30 ppm (b) [20,21]. For HACCu, compared to HACC, the new peaks appear at δ 8.12 (e,i), δ 6.96 (g,j), δ 6.62 (f,h) ppm respectively, which can be attributed to protons of coumarate anions. The new peaks of HACAs awared obvious at δ 4.41 ppm (e), δ 3.95 ppm (f), and δ 3.65 ppm (g) are attributed to the resonance of ascorbic acid anion. The signals at δ 6.13–7.40 ppm (e–k) are attributed to the protons of the *p*-coumaric acid. In particular, the signals at δ 7.40, δ 7.25, δ 6.51, and δ 6.13 ppm represent protons on H-f, H-g, H-e, and H-h, respectively. At the same time, for HACHy, HACSa, HACGa, new peaks appear at δ 6.3–7.6 ppm, δ 6.75–7.75 ppm, δ 7.01 ppm, which can be attributed to protons of benzene ring [19]. As for HACCa, the signals at δ 7.25, δ 6.21 ppm are related to the protons of H-h, H-i, respectively. Furthermore, in the range of δ 6.5–7.1 ppm is assigned to the protons on the benzene ring [3]. At the same time, the signal at δ 3.18 ppm which can be assigned to the resonance of hydrogen of trimethyl ammonium groups still exists in the spectra of conjugates of organic acid salt and hydroxypropyltrimethyl ammonium chitosan. Besides, the structures of chitosan derivatives are also confirmed by ^13^C NMR spectra. As shown in Figure 3, the specific chemical shifts of the carbons of acid anions are signed clearly. The peaks at δ 55.10–105.68 ppm obviously observed are attributed to the chemical shifts of chitosan. For the ^13^C NMR spectra of HACC, an obvious chemical shift at δ 57.25 ppm is assigned to the resonance of carbons of –N^+^(CH_3_)_3_. Meanwhile, the other signals for HACC can be well observed: δ = 69.19 ppm (c), δ = 64.81 ppm (b), δ = 51.85 ppm (a) [21,22]. As for conjugates of organic acid salt and hydroxypropyltrimethyl ammonium chitosan, compared to HACC, new signals appeared at δ 174.23, δ 176.18, δ 177.12, δ 176.57, δ 175.34, δ 176.93, δ 177.08 and δ 176.41 ppm are attributed to carbons of COO^−^ groups in HACCu, HACAs, HACFe, HACHy, HACSa, HACPc, HACCa, and HACGa [19,23]. Hence, these data sufficiently indicate that successful conjugations of organic acid salt and hydroxypropyltrimethyl ammonium chitosan.

#### 2.1.3. Elemental Analysis

The degrees of substitution (DS) was calculated by elemental analysis. The degree of substitution and the yields of chitosan derivatives are shown in Table 1. The degrees of substitution of HACC is the highest. Meanwhile, among all the conjugations of organic acid salt and hydroxypropyltrimethyl ammonium chitosan, HACPc presents the highest degrees of substitution.

### 2.2. Water Solubility

Figure 4 shows the solubility of chitosan derivatives (distilled water, 1.0 mg/mL) and chitosan (distilled water and 1% HAc). Obviously, chitosan is insoluble in distilled water, however, after quaternization, chitosan derivatives (HACC) is soluble in distilled water. It is also proved that quaternization is an important means to improve the water solubility of chitosan. In addition, all chitosan derivatives, including HACCu, HACAs, HACFe, HACHy, HACPc, HACSa, HACCa, HACGa, have excellent water solubility, and can be prepared to aqueous solution (0.1–1.0 mg/mL) at room temperature. The specific water solubility of chitosan and its derivatives is shown in Table 2. We used water-soluble low molecular chitosan in antioxidant and cytotoxicity assay because of the poor solubility of chitosan.

### 2.3. Antioxidant Activity

In this paper, the in vitro antioxidant activities, including DPPH-radical scavenging activity, superoxide-radical scavenging activity, hydroxyl-radical scavenging activity, and reducing power, were carried out to evaluate the antioxidant activities of ascorbic acid (AA), chitosan and chitosan derivatives. The antioxidant activities of chitosan and chitosan derivatives are shown in Figure 5, Figure 6, Figure 7 and Figure 8.

#### 2.3.1. Scavenging Ability of DPPH Radical

Figure 5 shows the DPPH-radical scavenging activity of chitosan, HACC, and target products. From the figure, we can find several conclusions as follows: firstly, chitosan and HACC have almost no DPPH-radical scavenging activity. However, the enhanced DPPH-radical scavenging activity of the conjugates of organic acid salt and hydroxypropyltrimethyl ammonium chitosan including HACCu, HACAs, HACFe, HACHy, HACSa, HACPc, HACCa, and HACGa are very evident. Secondly, for all the tested samples, with the augment of sample concentration, the scavenging effect of the conjugates of organic acid salt and hydroxypropyltrimethyl ammonium chitosan increases gradually. For instance, when the concentrations of HACGa are 0.1, 0.2, 0.4, 0.8, and 1.6 mg/mL, the scavenging effects are 62.13%, 74.81%, 79.08%, 88.54%, and 96.53% respectively. Thirdly, the order of scavenging effects are ranked as follows: HACGa > HACCa > HACAs > HACFe > HACPc > HACCu > HACSa > HACHy > HACC > CS.

#### 2.3.2. Scavenging Ability of Superoxide Radical

The superoxide-radical scavenging activity of chitosan and chitosan derivatives was also evaluated in Figure 6. As from the Figure 6, it can be seen that the scavenging activity against superoxide-radical of HACGa, HACCa, HACAs, HACFe, HACCu, HACSa, HACPc, and HACHy have been improved obviously comparing to chitosan and HACC. Specifically, the scavenging activity increases with the augment of sample concentration. Furthermore, after the introduction of organic acid salt, the conjugates of organic acid salt and hydroxypropyltrimethyl ammonium chitosan have a particularly well scavenging activity. Such as HACGa, HACAs, and HACCa, their scavenging effect achieves 100% at a concentration of 1.6 mg/L.

#### 2.3.3. Scavenging Ability of Hydroxyl Radical

The hydroxyl-radical scavenging activity of ascorbic acid, chitosan and chitosan derivatives are shown in Figure 7. The hydroxyl radicals are produced by the following Fenton reaction (Fe^2+^ + H_2_O_2_ = Fe^3+^ + •OH + OH^−^). The hydroxyl radicals will oxidize Fe^2+^ to Fe^3+^, and only the Fe^2+^ can combine with safranine O to form a red complex. The degree of discoloration can reflect the scavenging activity. As shown in the figure, it is found that the hydroxyl-radical scavenging activities of all tested samples are positive correlated with the concentrations. Meanwhile, chitosan and HACC showed relatively weak hydroxyl-radical scavenging activity. After modification, the conjugates of organic acid salt and hydroxypropyltrimethyl ammonium chitosan possessed much stronger radical scavenging ability compared with chitosan and HACC.

#### 2.3.4. Reducing Power

The antioxidant activity of chitosan and chitosan derivatives was also evaluated using reducing power assay and the result is presented in Figure 8. As shown in Figure 8, taking the reducing power of ascorbic acid as positive control, chitosan and HACC possesses the weak reducing power, and their absorbances are only 0.5 and 1.1 at the concentration of 1.6 mg/L. Meanwhile, compared to chitosan and HACC, the conjugates of organic acid salt and hydroxypropyltrimethyl ammonium chitosan including HACGa, HACqCa, HACAs, HACFe, HACCu, HACSa, HACPc, and HACHy have relatively strong reducing power. In addition, as previously described, the order of reducing power is HACGa > HACCa > HACAs > HACFe > HACPc > HACCu > HACSa > HACHy > HACC > CS.

The results showed that hydroxypropyltrimethyl ammonium chitosan organic acid salt including HACGa, HACCa, HACAs, HACFe, HACCu, HACSa, HACPc, and HACHy, had much stronger antioxidant activity. In summary, considering the degree of substitution, the antioxidant activities of chitosan derivatives are ranked as follows: HACGa > HACCa > HACAs > HACFe > HACPc > HACCu > HACSa > HACHy > HACC > CS. Specifically speaking, the conclusions were as follows: firstly, the conjugates of organic acid salt and hydroxypropyltrimethyl ammonium chitosan possessed strong antioxidant activity, which can confirm that the introduction of organic acid salt obviously enhanced the antioxidant activity of chitosan. This is related to the density of positive charge. Specifically, the antioxidant activity is associated with the density of positive charge, as the positive charge could attract the single electron of free radicals to inhibit the free radical chain reaction. Secondly, the antioxidant activity of chitosan derivatives varies with the quantities of phenolic hydroxyl [15]. In particular, HACGa, HACCa, HACPa, and HACSa had the same types of phenolic hydroxyl (–OH), and they exhibited different antioxidant activities due to the different amounts of phenolic hydroxyl (–OH). That was to say the antioxidant activities would enhance with the amounts of phenolic hydroxyl (–OH) [19]. In the mechanism of action of phenol antioxidants, terminating the chain reaction is the most important link. Because phenolic hydroxyl and radical can react to produce more stable semi-quinone radical. Hence, the antioxidant activities of the target products showed the order of HACGa > HACCa > HACPa > HACSa.

### 2.4. Cytotoxicity Assay

The cytotoxicity of chitosan and its derivatives was studied by CCK-8 assay to characterize its biocompatibility. After culture for 24 h, the cell viability of L929 cells, which were cultured by chitosan and its derivatives at series of concentrations, was shown in Figure 9. At the tested concentration, the cell viabilities of all the samples exceeded 80%. This value indicated that the growth inhibition of the samples on the cell was weak enough to be ignored, thus the products had good biocompatibility. Moreover, samples such as HACCu, HACPc, and HACCa exhibited no growth inhibition effect against L929 cells at the test concentrations. Specially, it was found that the cell viability of HACCu even exceeded 100%, and the values were positive correlated with the concentrations, which indicated a promotion effect of the cell. The result further proved the biocompatibility of the products. Therefore, chitosan derivatives could be considered to have good biocompatibility because of their low cytotoxicity.

## 3. Materials and Methods

### 3.1. Materials

Chitosan (MW 200 kDa, the degree of deacetylation 85%) was purchased from Qingdao Baicheng Biochemical Corp. (Qingdao, China). The materials, including 3-chloro-2-hydroxypropyltrimethyl ammonium chloride, sodium hydroxide, cumaric acid, ascorbic acid, ferulic acid, *p*-coumaric acid, caffeic acid, gallic acid, salicylic acid, hydrochloric acid, and hydroxybenzoic acid were purchased from Sigma-Aldrich Chemical Corp. (Shanghai, China). Sodium hydroxide, isopropyl alcohol, and ethanol were provided by Sinopharm Chemical Reagent Co., Ltd. (Shanghai, China). All the chemical solvents and reagents were obtained from commercial sources and used as received.

### 3.2. Analytical Methods

#### 3.2.1. Fourier Transform Infrared (FTIR) Spectroscopy

The FTIR spectra of chitosan and chitosan derivatives were recorded on a Jasco-4100 FTIR spectrometer (Tokyo, Japan), provided by JASCO China Co. Ltd. (Shanghai, China) at a resolution of 4.0 cm^−1^ in the 4000–400 cm^−1^ range. The tested samples were mixed with KBr disks and scanned 32 times at 25 °C for measurement.

#### 3.2.2. Nuclear Magnetic Resonance (NMR) Spectroscopy

The ^13^C NMR spectra and ^1^H NMR spectra were recorded by Bruker Avance III 500 NMR Spectrometer (500 MHz, Fällanden, Switzerland, provided by Bruker Tech. and Serv. Co., Ltd., Beijing, China) at 25 °C. The solvents of all tested samples were D_2_O.

#### 3.2.3. Elemental Analysis

Element analysis (C, H, N) was determined by Vario Micro Elemental Analyzer (Elementar, Hanau, Germany), and on the basis of the carbon-nitrogen ratios, the degrees of substitution (DS) of all tested chitosan derivatives can be calculated [23]. The equation was as follows:(1)$$\;\;\;\;\;DS_{1}=\frac{n_{1}\times M_{C}-M_{N}\times W_{C/N}}{n_{2}\times M_{C}}$$
(2)$$DS_{2}=\frac{n_{1}*M_{C}-M_{N}\times W_{C/N}-n_{2}\times M_{C}\times DS_{1}}{M_{N}\times W_{C/N}-n_{3}\times M_{C}}$$
(3)$$DS_{3}=\frac{M_{N}\times W_{C/N}+M_{N}\times W_{C/N}\times DS_{2}-n_{1}\times M_{C}\times DS_{2}-n_{2}\times M_{C}\times DS_{1}}{n_{4}\times M_{C}}$$
where *DS*_1_, *DS*_2_, *DS*_3_ represents the deacetylation degree of chitosan, the degrees of substitution of HACC, and the degrees of substitution of organic acid salt-hydroxypropyltrimethyl ammonium chitosan conjugates. *M**_N_* and *M**_C_* are the molar masses of nitrogen and carbon, *M_C_* = 12, *M_N_* = 14, respectively; *n*_1_, *n*_2_, *n*_3_, *n*_4_ are the number of carbons of chitin, acetamido group, 2-chloromethyl ammonium chloride, and organic acid group, *W_C/N_* means the mass ratio between carbon and nitrogen in chitosan derivatives.

### 3.3. Synthesis of Chitosan Derivatives

The organic acid-hydroxypropyltrimethyl ammonium chitosan conjugates were synthesized by multistep reactions (Scheme 1). Firstly, HACC was synthesized according to the previous method [24]. Chitosan (0.16 g, 1 mmol) was dissolved in isopropanol ahead of adding 0.4 mL 40% (*w/v*) NaOH aqueous solution, and the mixture was stirred at the speed of 3500 rpm/min at 60 °C for 4 h. Whereafter, 1.25 mL 60% (*w/v*) 3-chloro-2-hydroxypropyl trimethyl ammonium chloride solution was dropped into the front mixture with stirring at the speed of 3500 rpm/min at 80 °C. After 10 h of reaction, appropriate amount of hydrochloric acid added to ensure the pH of mixture was adjusted to 7. Then substantial anhydrous ethanol was used to precipitate. The precipitation (HACC) was rinsed with 80% ethanol, and dried in vacuo. After that, HACC (0.187 g, 0.6 mmol) was dissolved in 20% organic acid salt solution in order to replace the chloride ions with organic radical ions. After dialyzing with distilled water for 48 h, the solution was dried in vacuo for the purpose of obtaining chitosan derivatives bearing organic acid salt based hydroxypropyltrimethyl ammonium chitosan.

### 3.4. Water Solubility Assay

The water solubility was measured by following method [25]. The chitosan and chitosan derivatives (0.5 g) were dispersed in 0.5 mL distilled water, and the mixture was stirred at room temperature until dissolution equilibrium. After centrifugation and filtration, the undissolved solids were rinsed with ethanol, and dried in vacuo overnight. The water solubility was measured by the following equation:(4)$$\mathrm{Water}\;\mathrm{solubility}=\frac{500-m_{1}}{0.5},$$
where *m*_1_ is the weight of the undissolved solids (mg).

### 3.5. Antioxidant Activity Assay

#### 3.5.1. DPPH-Radical Scavenging Activity Assay

The DPPH radical scavenging ability was calculated by the following method [26]. Firstly, all the tested samples with original concentration of 10 mg/mL (0.03, 0.06, 0.12, 0.24, and 0.48 mL) and 2 mL ethanol solution of DPPH (180 µM) were incubated for 30 min. Meanwhile, in the control group, DPPH was replaced by 2 mL anhydrous ethanol, and in the blank, samples were instead of distilled water. Then, the absorbance of rest DPPH radical was measured at 517 nm against a blank. Three replicates for every sample concentration were measured and the scavenging effect was calculated by the following equation:(5)$$\mathrm{Scavenging}\;\mathrm{effect}\;\left(\%\right)=\left[1-\frac{A_{sample\;517\;nm}-A_{control\;517\;nm}}{A_{blank\;517\;nm}}\right]\times 100,$$
where *A_sample_*
_517*nm*_, *A_control_*
_517 *nm*_, and *A_blank_*
_517 *nm*_ represent the absorbance of the sample, the control (ethanol instead of DPPH for each concentration), and the blank, respectively.

#### 3.5.2. Superoxide-Radical Scavenging Activity Assay

According to Tan [16], the superoxide-radical scavenging activity of the tested samples were calculated by follows: the reaction system including chitosan or chitosan derivatives (5 mg/mL, 0.06, 0.12, 0.24, 0.48, and 0.96 mL), phenazine methosulfate (PMS, 30 μM), nicotinamide adenine dinucleotide reduced (NADH, 338 μM), and nitro blue tetrazolium (NBT, 72 μM) in Tris-HCl buffer (16 mM, pH 8.0), was incubated for 5 min at room temperature. The absorbance was recorded at 560 nm, repeated three times and averaged.

The formula of superoxide-radical scavenging activity is as follows:(6)$$\mathrm{Scavenging}\;\mathrm{effect}\;\left(\%\right)=\left[1-\frac{A_{sample\;560\;nm}-A_{control\;560\;nm}}{A_{blank\;560\;nm}}\right]\times 100,$$
where *A_sample_*
_560 *nm*_*, A_control_*
_560 *nm*_*, A_blank_*
_560 *nm*_ are the absorbance of the samples, the control (NADH was substituted with distilled water), and the blank (samples were substituted with distilled water), respectively.

#### 3.5.3. Hydroxyl-Radical Scavenging Activity Assay

According to Saravanakumar et al. [26], the hydroxyl-radical scavenging ability was calculated by follows: the total reaction system of 4.5 mL, including the tested samples of chitosan or chitosan derivatives (10 mg/mL, 0.045, 0.09, 0.18, 0.36, and 0.72 mL), EDTA-Fe^2+^ (220 µM), safranine O (0.23 µM), and H_2_O_2_ (60 µM) in potassium phosphate buffer (150 mM, pH = 7.4), was incubated at 37 °C for 30 min. Then, the absorbance of the reaction system was measured at 520 nm. Three replicates for each sample were tested and the hydroxyl-radical scavenging activity was calculated via the following equation:(7)$$\mathrm{Scavenging}\;\mathrm{effect}\;\left(\%\right)=\frac{A_{sample\;520\;nm}-A_{blank\;520\;nm}}{A_{control\;520\;nm}-A_{blank\;520\;nm}}\times 100,$$
where *A_sample_*
_5__2__0 *nm*_*,*
*A_control_*
_5__2__0 *nm*_*,*
*A_blank_*
_5__2__0 *nm*_ represent the absorbance of the samples, the control (H_2_O_2_ was substituted with potassium phosphate), and the blank (the samples were substituted with distilled water), respectively.

#### 3.5.4. Reducing Power Assay

The reducing power was measured by the following method [16]: firstly, the reaction mixture including chitosan or chitosan derivatives with a series of concentrations (0.1, 0.2, 0.4, 0.8, 1.6 mg/mL), potassium ferricyanide(1%, *w/v*) dissolved in sodium phosphate buffer (0.2 M, pH = 6.6), was reacted at 50 °C. Twenty minutes later, 1.0 mL of trichloroacetic acid (10%, *w/v*) was added in order to terminate the reaction. After centrifugation, 1.2 mL of deionized water and 0.3 mL of ferric chloride solution (0.1%, *w/v*) were added. After 10 min, the absorbance was measured at 700 nm, repeated three times and averaged. The reducing power can be calculated by following equation.
Reducing power (%) = *A*_1 700 *nm*_*– A*_0 700 *nm*_(8)
where *A*_1 700 *nm*_*, A*_0 700 *nm*_ represent the absorbance of the samples and the blank.

### 3.6. Cytotoxicity Assay

The cytotoxicity of chitosan and its derivatives on L929 cells at a series of concentrations was measured in vitro according to a cell counting kit-8 (CCK-8) assay [7,27,28]. After culturing in RPMI medium (including 1% mixture of penicillin & streptomycin and 10% fetal calf serum) at 37 °C, the L929 cells were transferred to 96-well flat-bottom culture plates at a density of 1.0 × 10^5^ cells and incubated at 37 °C (5% CO_2_). All the samples at a series of concentrations (10.0, 100.0, 500.0, and 1000.0 μg/mL) were introduced into the cells which attached for 24 h. Then, the cells were cultured for 24 h. Whereafter, CCK-8 solution (10 μL) was added in each well. After incubation for 4 h at 37 °C, the absorbance at 450 nm was recorded by a microplate reader. Cell viability was measured by the following equation:(9)$$\mathrm{Cell}\;\mathrm{viability}\;\left(\%\right)=\frac{A_{sample\;450\;nm}-A_{blank\;450\;nm}}{A_{control\;450\;nm}-A_{blank\;450\;nm}}\times 100,$$
where *A_sample_*, *A_blank_*, and *A_control_* are the absorbance of the samples (including sample solution, cells, and CCK-8 solution), the blank (containing RPMI medium and CCK-8 solution), and the control (containing CCK-8 solution and cells).

### 3.7. Statistical Analysis

All the experiments were repeated three times and the data were expressed as mean ± the standard deviation (SD, *n* = 3). Significant difference analysis was determined using Scheffe’s multiple range test. A level of *p* < 0.05 was considered statistically significant.

## 4. Conclusions

In this research, we modified hydroxypropyltrimethyl ammonium chloride chitosan with eight different organic acid salt via ion exchange to acquire water-soluble and strong antioxidant activity chitosan derivatives. The in vitro antioxidant activities, including DPPH-radical scavenging activity, superoxide-radical scavenging activity, hydroxyl-radical scavenging activity, and reducing power, were carried out to evaluate the antioxidant activities of chitosan and chitosan derivatives after the structural characterization. The results indicated that the eight final chitosan derivatives exhibited higher antioxidant activities than chitosan and HACC. The results of this study showed the key role of phenolic hydroxyl in enhancing the antioxidant activities of chitosan. Furthermore, the cytotoxicity of chitosan and its derivatives was studied by CCK-8 assay to characterize their biocompatibility. The result further proved the biocompatibility of the products. So, this study provides a new method for the preparation of innocuous and effective antioxidant, and these new kinds chitosan derivatives represent a novel class of highly effective active polymers and may serve as antioxidant agents to partially substitute for the antioxidants in fields of food and pharmaceutical industry. The use of chitosan derivatives in pharmaceutical and food industry is an innovative, green and environmentally friendly technology that could be easily implemented at an industrial scale with little additional costs.

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
