# Peer review of "Modification of Hydroxypropyltrimethyl Ammonium Chitosan with Organic Acid: Synthesis, Characterization, and Antioxidant Activity"

_polymers, 2020, doi:10.3390/polym12112460_

Round 1

Reviewer 1 Report

It would be beneficial to more specifically formulate the aim of research at the end of Introduction.

In line 348 in equation for calculation of cell viability was written SCAVENGING EFFECT

In fig.8 the cytotoxicity of chitosan on L929 cells is too high if compared for example with recent publication of your research group in

Zhang et  al.Synthesis and Characterization of N,N,N-trimethyl-O-(ureidopyridinium)acetyl Chitosan Derivatives with Antioxidant and Antifungal Activities. MARINE DRUGS,2020,18, 163

At concentration 0 mg/ml cell viability must be 100 percent for all samples, but cell viability at 0 concentration in FIG.8 is lower for a number of samples

Author Response

Responds to reviewer 1:

  1. It would be beneficial to more specifically formulate the aim of research at the end of Introduction.

Answer: Thank you for your kind suggestions and according to your recommendation, we have formulated the aim of research at the end of Introduction. Based on the above, we hope to obtain a kind of novel chitosan derivatives with good bioactivity, biocompatibility, and degradability, which will be widely used as an antifungal agent or antioxidant in food, medicine, cosmetics, and other fields (Lines 70-72). Thank you for your kind suggestion.

  1. In line 348 in equation for calculation of cell viability was written SCAVENGING EFFECT.

Answer: Thank you for your kind suggestions and according to your recommendation, we have corrected in the revised paper (Line 365). Very sorry for this mistake.

  1. In fig.8 the cytotoxicity of chitosan on L929 cells is too high if compared for example with recent publication of your research group in Zhang et al. Synthesis and Characterization of N,N,N-trimethyl-O-(ureidopyridinium)acetyl Chitosan Derivatives with Antioxidant and Antifungal Activities. MARINE DRUGS,2020,18, 163. At concentration 0 mg/ml cell viability must be 100 percent for all samples, but cell viability at 0 concentration in Fig.8 is lower for a number of samples.

Answer: Thank you for your kind suggestions. The cytoxicity of chitosan on L929 cells was related to the molecular weight of chitosan. The chitosan used in the paper of zhang et al. was a high molecular weight chitosan. However, in this paper, We used water-soluble low molecular chitosan in antioxidant and cytotoxicity assay because of the poor solubility of chitosan. According to your recommendation, we illustrated this in the water solubility section (Lines 168-169). At the same time, there is experiment error in our experiment. Although all the experiments were performed in triplicate and the data were expressed as mean ± the standard deviation, there is still some error in our experiment, so the cell viability at 0 concentration in Fig.8 is lower for a number of samples. Thank you for your kind suggestions and we hope meet with approval.

Reviewer 2 Report

The current manuscript submitted by Dr Guo et al describes the “Synthesis of chitosan derived analogs and their antioxidant activity in vitro”. Introduction part in the current manuscript was well documented by the authors. The synthetic method has been already published in their prior publication. However, the authors explored their idea towards the preparation of new analogs and studied their in vitro antioxidant activity. Some of the synthesized analogs have shown good antioxidant activity. This work will be of interest to others in this field and hence I support for publication in “Polymers”. However, some typo errors have been found as well as the authors need to be addressed the following concerns before a consideration for acceptance in “Polymers”.

 Comments:

  1. Page 2, line 52, replace ‘sufficient’ with ‘significant’.
  2. Please make it more consistence through the manuscript about the cytotoxicity of all tested compounds, whether it is ‘low’ or ‘no cytotoxicity’.?
  3. Page 2, line 91, delete ‘And’.
  4. I advise the authors to integrate the proton signals and mention the no. of protons in the 1H-NMR spectra, Fig 2.
  5. What does mean stands for ‘VC’ in Fig 4-7? Please abbreviate it.
  6. Page 13, line 348, please recheck the equation again, whether it is cytotoxicity or scavenging effect..?
  7. How the authors confirmed all the synthesized target compounds have good water solubility.?
  8. Page 13, line 365, replace ‘characterize its’ with ‘characterize their’.
  9. Finally, English needs thorough revision throughout the text by a native speaker.

In summary, current manuscript requires a minor revision before a consideration for acceptance.

Author Response

Responds to reviewer 2:

The current manuscript submitted by Dr Guo et al describes the “Synthesis of chitosan derived analogs and their antioxidant activity in vitro”. Introduction part in the current manuscript was well documented by the authors. The synthetic method has been already published in their prior publication. However, the authors explored their idea towards the preparation of new analogs and studied their in vitro antioxidant activity. Some of the synthesized analogs have shown good antioxidant activity. This work will be of interest to others in this field and hence I support for publication in “Polymers”. However, some typo errors have been found as well as the authors need to be addressed the following concerns before a consideration for acceptance in “Polymers”. In summary, current manuscript requires a minor revision before a consideration for acceptance.

Comments:

  1. Page 2, line 52, replace ‘sufficient’ with ‘significant’.

Answer: Thank you for your kind suggestions and according to your recommendation, we have replaced ‘sufficient’ with ‘significant’, very sorry for this mistake.

  1. Please make it more consistence through the manuscript about the cytotoxicity of all tested compounds, whether it is ‘low’ or ‘no cytotoxicity’.?

Answer: Thank you for your kind suggestions and according to your recommendation, we have unified the description of the cytotoxicity of all tested compounds (Lines 25, 89, 257), very sorry for this mistake.

  1. Page 2, line 91, delete ‘And’.

Answer: Thank you for your kind suggestions and according to your recommendation, we have deleted “And”, very sorry for this mistake.

  1. I advise the authors to integrate the proton signals and mention the no. of protons in the 1H-NMR spectra, Fig 2.

Answer: Thank you for your kind suggestions and according to your recommendation, we have divided the protons signals and corresponded to their positions in the Fig. 2. Meanwhile, we described the proton signals in the paper (Lines 129-136). In addition, the degrees of substitution of chitosan derivatives were calculated by elemental analysis, so we did not integrate hydrogen protons. Thank you for your kind suggestions and we hope meet with your approval.

  1. What does mean stands for ‘VC’ in Fig 4-7? Please abbreviate it.

Answer: Thank you for your kind suggestions and according to your recommendation, we have abbreviated “ascorbic acid” as “AA” (Line 176), and corrected the figures (Fig 5-8). Thank you for your kind suggestions and we hope meet with your approval.

  1. Page 13, line 348, please recheck the equation again, whether it is cytotoxicity or scavenging effect..?

Answer: Thank you for your kind suggestions and according to your recommendation, we have corrected in the revised paper (Line 365). Very sorry for this mistake.

  1. How the authors confirmed all the synthesized target compounds have good water solubility.?

Answer: Thank you for your kind suggestions and according to your recommendation, we have tested water solubility of chitosan and chitosan derivatives in order to confirm all the synthesized target compounds have good water solubility (Line 160-172, Lines 304-311). Thank you for your kind suggestions and we hope meet with approval.

  1. Page 13, line 365, replace ‘characterize its’ with ‘characterize their’.

Answer: Thank you for your kind suggestions and according to your recommendation, we have replaced ‘characterize its’ with ‘characterize their’. Very sorry for this mistake.

  1. Finally, English needs thorough revision throughout the text by a native speaker.

Answer: Thank you for your kind suggestions and according to your recommendation, we have checked grammar, verb tense of the manuscript and modified the sentences. Thank you for your kind suggestions and we hope meet with approval.